# Metabolic Engineering of *Pichia pastoris* for the Production of Triacetic Acid Lactone

**DOI:** 10.3390/jof9040494

**Published:** 2023-04-20

**Authors:** Linjuan Feng, Junhao Xu, Cuifang Ye, Jucan Gao, Lei Huang, Zhinan Xu, Jiazhang Lian

**Affiliations:** 1Key Laboratory of Biomass Chemical Engineering of Ministry of Education, College of Chemical and Biological Engineering, Zhejiang University, Hangzhou 310027, China; 2ZJU-Hangzhou Global Scientific and Technological Innovation Center, Zhejiang University, Hangzhou 311215, China

**Keywords:** tritactic acid lactone (TAL), acetyl-CoA, phosphoketolase/phosphotransacetylase pathway, xylose utilization, methanol biotransformation, *Pichia pastoris*

## Abstract

Triacetic acid lactone (TAL) is a promising renewable platform polyketide with broad biotechnological applications. In this study, we constructed an engineered *Pichia pastoris* strain for the production of TAL. We first introduced a heterologous TAL biosynthetic pathway by integrating the 2-pyrone synthase encoding gene from *Gerbera hybrida* (*Gh2PS*). We then removed the rate-limiting step of TAL synthesis by introducing the posttranslational regulation-free acetyl-CoA carboxylase mutant encoding gene from *S. cerevisiae* (*ScACC1**) and increasing the copy number of *Gh2PS*. Finally, to enhance intracellular acetyl-CoA supply, we focused on the introduction of the phosphoketolase/phosphotransacetylase pathway (PK pathway). To direct more carbon flux towards the PK pathway for acetyl-CoA generation, we combined it with a heterologous xylose utilization pathway or endogenous methanol utilization pathway. The combination of the PK pathway with the xylose utilization pathway resulted in the production of 825.6 mg/L TAL in minimal medium with xylose as the sole carbon source, with a TAL yield of 0.041 g/g xylose. This is the first report on TAL biosynthesis in *P. pastoris* and its direct synthesis from methanol. The present study suggests potential applications in improving the intracellular pool of acetyl-CoA and provides a basis for the construction of efficient cell factories for the production of acetyl-CoA derived compounds.

## 1. Introduction

The ever-increasing demand for natural resources and concerns over climate change have ignited a broad interest in the use of microbial cell factories for producing biofuels and chemicals [1]. Metabolic engineering allows for the manipulation of central carbon metabolic networks to produce specific target compounds [2], which offers advantages such as being environmentally-friendly, generating fewer by-products, and reducing production costs. Therefore, the development of an industrial platform for chemical biosynthesis through metabolic engineering is crucial in meeting the demands of sustainable development [3,4].

*Pichia pastoris* (also known as *Komagataella phaffii*), a yeast strain with a Generally Regarded as Safe (GRAS) status [5,6], has become increasingly popular due to its advantages for use as a versatile yeast cell factory, including high protein secretion, low glycosylation levels, a Crabtree negative phenotype, robustness in high-density fermentation, and high methanol tolerance [5,7]. The availability of genetic manipulation tools [8], such as the CRISPR/Cas9 system, has allowed for effective editing and genetic manipulation of the *P. pastoris* genome [9]. This advancement has enabled further research in metabolic flux regulation and metabolic engineering, in order to attain the desired metabolic flow to synthesize target metabolites [10].

Triacetic acid lactone (6-methyl-4-hydroxy-2-pyrone, TAL), one of the simplest polyketides synthesized by 2-pyrone synthase from *Gerbera hybrida* (Gh2PS), is a potential renewable platform compound and can be further converted into high value-added chemicals, such as additives, fragrances, and pharmaceuticals. TAL has a broad spectrum of applications in the chemical, material, food, and pharmaceutical industries [11,12]. Metabolic engineers have established a number of TAL-producing strains, including but not limited to *E. coli*, *S. cerevisiae*, and *Y. lipolytica*. By engineering the Gh2PS enzyme, TAL titer in *E. coli* reached up to 16.4 ± 0.5 mM in LB with 220 mM glycerol medium, with a TAL yield of 0.102 g/g glycerol [13]. However, low tolerance limited *E. coli* as a promising TAL-producing host. TAL production by industrial *S. cerevisiae* via fed-batch cultivation with ethanol feed was reported to be 5.2 g/L, which is considered to be limited by the availability of acetyl-CoA pool in this conventional organism [14]. *Y. lipolytica*, an oleaginous yeast, has been regarded as an attractive industrial workhorse for efficient TAL biosynthesis owing to its high flux through the key precursors, acetyl-CoA and malonyl-CoA. In a recent study, heterologous expression of *Gh2PS* along with cytosolic expression of alternative acetyl-CoA pathways were employed to improve TAL synthesis, and the best engineered *Y. lipolytica* strain resulted in a maximum titer of ~35.9 g/L in a bioreactor fermentation, with a yield up to 43% of the theoretical value from glucose [15]. 

To date, there has been no reported TAL synthesis in *P. pastoris*, especially through the assimilation of one-carbon compounds (e.g., methanol). Previous studies showed that the availability of the intracellular acetyl-CoA pool largely limited the synthesis of acetyl-CoA derivatives. To achieve efficient biotransformation of target metabolites, it is necessary to rewire the intracellular acetyl-CoA metabolic pathway [16]. As the metabolic network of acetyl-CoA in eukaryotes is tightly regulated and highly compartmentalized [17], the introduction of orthogonal cytosolic acetyl-CoA pathways into the host is a generally effective strategy to improve the overall pool of intracellular acetyl-CoA. Engineering efforts to improve intracellular synthesis of acetyl-CoA has been studied in several hosts, including *E. coli* [18], *S. cerevisiae* [19], *R. toruloides* [20], and *Y. lipolytica* [21]. The xylose-5-phosphate (Xu5P) specific phosphokinase/phosphoketolase pathway (PK pathway) can effectively synthesize cytosolic acetyl-CoA [22] from Xu5P, an intermediate of the oxidative phosphorylation pathway (oxPPP), through a two-step reaction without carbon loss [23]. This pathway has been employed to improve intracellular acetyl-CoA pool in several engineered strains [24,25]. 

In this study, we aimed to engineer *P. pastoris* for efficient production of TAL (Figure 1). We employed multiple metabolic engineering strategies to boost the supply of acetyl-CoA in *P. pastoris*, including the integration of PK pathway as well as the enhanced synthesis of the precursor Xu5P through the heterologous xylose utilization pathway and endogenous methanol utilization pathway. Our results demonstrated the potential of *P. pastoris* for the production of value-added acetyl-CoA derivatives from renewable resources, such as xylose and methanol.

## 2. Materials and Methods

### 2.1. Strains and Reagents

*P. pastoris* GS115-*Cas9* (*his4*::*Cas9*) constructed in previous studies was used as the parent strain [9]. Restriction enzymes and T4 DNA ligase were purchased from NEB (Ipswish, MA, UK). Phanta DNA polymerase and 2 × Taq PCR mix were purchased from Vazyme (Nanjing, China). DNA gel purification kit and plasmid extraction kit were purchased from Sangon Biotech (Shanghai, China). TAL standard was purchased from TCI (Shanghai, China). All chemicals were purchased from Sangon Biotech (Shanghai, China) unless stated otherwise. 

### 2.2. Plasmid Construction

The integration donor helper plasmids were constructed in our previous studies [9], containing an upstream and a downstream homologous arm of ~500 bp as well as the *pTEF1-Hind*III*-Nde*I*-t0547*, *pTEF1*-*Hind*III*-Nde*I-*tAOX1* and/or *pGAP*-*Ava*I-*Bgl*II-*tAOX1* cassettes. The helper plasmid HZP/HGP/HHP-sgRNA-IntX is composed of universal skeleton, resistance gene expression cassette, sgRNA scaffold and two *Bsa*I restriction sites. Z represented zeocin resistance marker, G represented G418 resistance marker, and H represented hygromycin B resistance marker. P represents plasmid. PCR-amplified genes sequences and sgRNA sequences were cloned into corresponding restriction enzyme sites by Gibson Assembly or the one-step cloning method. The gene encoding 2-Pyrone synthase *Gh2PS*, xylulose-5-phosphate specific phosphoketolase encoding gene *xPK*, and phosphotransacetylase encoding gene *PTA* were from *Gerbera hybrida*, *Leuconostoc mesenteroides*, and *Clostridium kluyveri*, respectively. Xylose reductase encoding gene *XR*, xylitol dehydrogenase encoding gene *XDH*, and xylulokinase encoding gene *XKS* were from *Scheffersomyces stipitis*. All the heterogeneous genes mentioned above were codon-optimized and chemically synthesized. Acetyl-CoA carboxylase encoding gene *ScACC1** was PCR-amplified from *S. cerevisiae* genome with two mutations (Ser659A and Ser1157A) to minimize SNF1-mediated protein degradation [26]. Six *Gh2PS* integration helper plasmids with different genomic integration loci were constructed for multi-copy integration of *Gh2PS* (Int1-*Gh2PS*-donor, Int11-*Gh2PS*-donor, Int20-*Gh2PS*-donor, Int32-*Gh2PS*-donor, Int33-*Gh2PS*-donor, and Int34-*Gh2PS*-donor). Int39-*ScACC1**-donor was constructed for the integration of *ScACC1** at Int39. Int56-*XR-XDH*-donor was constructed for the integration of *XR* and *XDH* at Int56. Int1*-XKS*-donor was constructed for the integration of *XKS* at Int1. Int35-*xPK-*donor and Int59*-PTA*-donor were constructed for the integration of *xPK* and *PTA* at Int35 and Int59, respectively. 

The gene deletion helper plasmids included a universal skeleton, an upstream and a downstream homologous arm sequences of ~500 bp. PCR-amplified homologous arm sequences were pieced together using the Gibson assembly method. Det*PFK1*-donor and Det*PYK1*-donor were constructed for the deletion of *pfk1* and *pyk1*, respectively. The helper plasmid HZP-gRNA-Det*PFK1*/Det*PYK1* included a universal skeleton, zeocin resistance gene expression cassette, sgRNA scaffold and 20 bp sgRNA sequence corresponding to *pfk1*/*pyk1* gene. 

All plasmids constructed in this study are listed in Appendix A. All primers synthesized by Youkang Biotechnology Co., Ltd. (Hangzhou, China) are listed in Appendix A. All heterologous genes used in this study were synthesized by GenScript Biotech (Nanjing, China) and listed in Appendix A.

### 2.3. Yeast Strain Construction 

The CRISPR/Cas9 system was used for genetic manipulation in yeast (Appendix A) [9]. The PCR-amplified gene integration/deletion fragments from helper plasmids (~1000 ng) and the corresponding sgRNA plasmids (~500 ng) were transformed into *P. pastoris* competent cells by the Lin–Cereghine electro-transformation method [27]. The transformants were verified by diagnostic PCR and DNA sequencing. All strains used in this study are listed in Table 1.

### 2.4. Medium and Cultivation

*E. coli* DH5α, used for recombinant DNA manipulation, was cultured at 37 °C in LB broth or on agar plates with 100 mg/L ampicillin. Yeast strains were routinely cultured at 30 °C in YPD medium (10 g/L yeast extract, 20 g/L peptone, and 20 g/L glucose) or SCD/SCX/SCM medium (3.4 g/L yeast nitrogen base, 5 g/L ammonium sulfate, and 20 g/L glucose, xylose, or methanol, respectively). A total of 20 g/L agar was added to prepare solid media. Zeocin, Hygromycin B, and G418 were added with a final concentration of 100 mg/L, 200 mg/L, and 200 mg/L, respectively, for the selection of engineered strains. 

For TAL production, single colonies were picked from a YPD plate and inoculated into 5 mL YPD liquid media, which were grown at 30 °C for 48 h. A seed culture of 500 μL was washed twice with the corresponding fermentation media and then inoculated into 25 mL fermentation media in 250 mL shake flasks with 250 rpm and 30 °C for 72 h. Samples were taken every 24 h to analyze biomass, sugar content, and TAL titer.

### 2.5. Analytical Methods

Cell growth was monitored by measuring the optical density at 600 nm (OD_600_) with a Synergy™ H1 Multi-Mode Microplate Reader (BioTek, Winooski, VT, USA). Fermentation broth was centrifuged at 14,000 rpm for 5 min and the supernatant was diluted 10~100-fold by ddH_2_O for the analysis of glucose, xylose, and TAL. Residual glucose and xylose were quantified by a SBA-90 biosensor (Shangdong Academy of Sciences, Jinan, China), while TAL was analyzed by HPLC (Agilent, Santa Clara, CA, USA) equipped with a C18 column (Agilent, USA) and a UV absorbance detector. The column was maintained at 35 °C with a flow rate of 0.6 mL/min for 16 min. TAL was detected at 280 nm with a gradient program, which was started with a mixture of 95% solvent A (0.1% acetic acid in ddH_2_O) and 5% solvent B (100% methanol), changed linearly to 75% solvent A and 25% solvent B over a period of 6.8 min, then shifted linearly to 5% solvent A and 95% solvent B in 1 min, and finally returned to the original composition of 95% solvent A and 5% solvent B at 16 min.

## 3. Results

### 3.1. Production of TAL in P. pastoris via the Introduction of 2-Pyrone Synthase 

TAL can be biosynthesized by a type III polyketide synthase (PKS) from *Gerbera hybrida* [28], 2-pyrone synthase (encoded by *Gh2PS*), using a starter acetyl-CoA and two extender malonyl-CoA molecules (Figure 1). To enable TAL synthesis in *P. pastoris*, we first introduced the codon-optimized *Gh2PS* gene into the strain GS115-*Cas9*. The resultant strain PpTAL1, with a single integration of *Gh2PS,* produced ~1.0 g/L TAL using shake flask fermentation in YPD medium (Figure 2 and Appendix A), indicating the potential of *P. pastoris* for the production of polyketides. 

### 3.2. Overexpression of ScACC1* and Multi-Copy Integration of Gh2PS to Enhance TAL Production

Previous studies have reported that boosting the level of malonyl-CoA, the direct precursor of TAL, can enhance TAL synthesis [29,30]. To investigate the impact of converting acetyl-CoA to malonyl-CoA on TAL production, we implemented two independent metabolic engineering strategies in this study: replacing the endogenous *ACC1* promoter with the stronger *GAP* promoter (PpTAL2), or introducing a posttranslational regulation-free *ACC1* mutant from *S. cerevisiae* (PpTAL3) [26]. Disappointingly, TAL titer of the strain PpTAL2 obtained through the first strategy showed no significant change, while TAL titer of the strain PpTAL3 (~1.1 g/L TAL) was 11% higher than that of the control strain (Figure 2), demonstrating that the *ACC1* mutant from *S. cerevisiae* functioned effectively to synthesize malonyl-CoA and was beneficial for TAL production in *P. pastoris*. In addition, to overcome the rate-limiting step in TAL synthesis, we increased the copy number of *Gh2PS* integrated into the genome. With more copies of *Gh2PS*, we observed gradually increased production of TAL fermented in YPD medium. The strain PpTAL8, with six copies of *Gh2PS*, was able to produce ~2.7 g/L TAL in shake flask fermentation (Figure 2), which was 2.7-fold higher than that of PpTAL1 (Figure 2). Considering that PpTAL6 with four copies of *Gh2PS* produced comparable amounts of TAL (2.4 g/L) and the growth of PpTAL7 and PpTAL8 was negatively affected by more copies of *Gh2PS*, we chose PpTAL6 for subsequent metabolic engineering studies. 

### 3.3. Introduction of PK Pathway to Boost Acetyl-CoA Supply

Acetyl-CoA is a key precursor involved in TAL synthesis. Previous studies have shown that the PK pathway is effective and ATP-costless for the synthesis of cytosolic acetyl-CoA [22], when compared with the endogenous PDH bypass pathway [31], converting pyruvate to acetyl-CoA through a three-step reaction sequentially catalyzed by pyruvate decarboxylase (PDC), acetaldehyde dehydrogenase (ALD), and acetyl-CoA synthetase (ACS) (Figure 1).

Thus, we further evaluated the PK pathway for enhancing the synthesis of cytosolic acetyl-CoA and accordingly the production of TAL. Specifically, we overexpressed the corresponding genes, *xPK* from *L. mesenteroides* and *PTA* from *C. kluyveri* in PpTAL6 to construct PpTAL9 and in PpTAL10 to construct PpTAL11. As synthetic medium with clear components is more suitable to evaluate the performance of PK pathway, we carried out all the subsequent engineering efforts in synthetic medium (e.g., SCD and SCX).

Unexpectedly, fermentation results showed no significant difference in TAL yield in PpTAL10 and PpTAL11 (Appendix A), indicating that either PK pathway had no intracellular functions or the supply of precursor (i.e., Xu5P) was insufficient.

### 3.4. Verification of PK Pathway in P. pastoris via Growth Complementation

As the PK pathway failed to increase the production of TAL, we then set out to verify the intracellular functions of the PK pathway, via growth complementation of the phosphofructokinase (*PFK1*) or pyruvate kinase (*PYK1*) deficient strain. PFK1 catalyzes the irreversible production of fructose-1,6-bisphosphate (FBP) from fructose-6-phosphate (F6P), and PYK1 catalyzes the formation of pyruvate (PYR) from phosphoenolpyruvate (PEP). We knocked out *PFK1* and *PYK1* individually in *P. pastoris* and found that the *pfk1Δ* or *pyk1Δ* strain failed to grow in SCD medium (Figure 3A), indicating that their endogenous metabolic pathway through glucose metabolism to synthesis of acetyl-CoA was blocked, while the normal growth of defective strains in ethanol indicated that *pfk1* or *pyk1* knock-out did not affect the utilization of ethanol (Figure 3B). After introducing the *xPK* and *PTA* into the *pfk1Δ* and *pyk1Δ* strains, the engineered strains *pfk1Δ*::*xPK/PTA* and *pyk1Δ*::*xPK/PTA* were constructed, and the growth recovery was achieved in SCD medium (Figure 3A). These results proved that PK pathway could bypass the endogenous glycolysis pathway and synthesize cytoplasmic acetyl-CoA from the intermediate xylose-5-phosphate (Xu5P) via the pentose phosphate pathway to recover cell growth.

### 3.5. Production of TAL from Xylose

After verification of the intracellular functions of the PK pathway, the failure to significantly increase TAL production could result from the insufficient supply of the direct precursor Xu5P. The xylose utilization pathway has been well established to assimilate xylose into Xu5P by a three-step enzyme reaction catalyzed by xylose reductase (XR), xylitol dehydrogenase (XDH), and xylulokinase (XKS) [32,33]. As the main hydrolysis product of hemicellulose and the second most abundant sugar present in nature after glucose, xylose is considered a promising renewable resource and a substantial alternative carbon source for the economical production of biofuels and chemicals [34]. However, natural *P. pastoris* lacks the ability to utilize xylose. Thus, three xylose assimilation-related genes, *XR*, *XDH*, and *XKS* from *Sc. stipitis* were cloned and integrated into GS115-*Cas9* and PpTAL6 to construct WT-XUP and PpTAL10, respectively. Our results showed that the cell growth of WT-XUP and PpTAL10 was significantly improved in SCX medium with xylose as the sole carbon source (Figure 4A), indicating a higher xylose utilization efficiency than that of the control strains. Growth on glucose was also tested as the positive control (Figure 4B). The highest OD_600_ of the strain WT-XUP and PpTAL10 in SCX was comparable to that of glucose. This demonstrated that the introduction of heterologous pentose metabolic pathway genes enabled efficient xylose utilization in *P. pastoris*. 

We further introduced the PK pathway into PpTAL10 to construct strain PpTAL11. As expected, the titer of TAL in PpTAL11 was 1.6-fold higher than that of PpTAL10, reaching up to 825.6 mg/L in SCX medium (Figure 4C). The glucose and xylose consumption profiles of PpTAL10 and PpTAL11 showed that xylose consumption was more efficient and resulted in higher cell densities than glucose (Figure 4D and Appendix A). These results indicated that the xylose utilizing strain effectively assimilated xylose into Xu5P, thereby increasing the carbon flux of the PK pathway for the synthesis of cytosolic acetyl-CoA, resulting in improved TAL production.

### 3.6. Production of TAL from Methanol 

In addition to the xylose utilization pathway, Xu5P is also an essential intermediate metabolite for methanol assimilation, employed through the xylulose monophosphate (XuMP) pathway located in the peroxisomes of *P. pastoris* [35]. Methanol is firstly oxidized to formaldehyde by alcohol oxidase (AOX), which is then condensed with Xu5P by dihydroxyacetone synthase (DAS) to form two central carbon intermediates, glyceraldehyde-3-phosphate (GAP) and dihydroxyacetone phosphate (DHAP) [36] (Figure 1). Thus, we combined the endogenous methanol assimilation with the PK pathway to improve TAL production. To test this approach, we fermented PpTAL9 in SC medium supplemented with 2% methanol (SCM). Our results showed that PpTAL9 successfully synthesized TAL using methanol as the sole carbon and energy source, with a titer of 57.1 mg/L (Figure 5), which was 2.8-fold higher than that of PpTAL6 (20.5 mg/L). TAL yield from methanol by PpTAL9 was 0.0010 g/g, representing 0.156% of the theoretical maximum yield. These results validated the effectiveness of the PK pathway in promoting acetyl-CoA synthesis with methanol as carbon source and demonstrated that *P. pastoris* could be harnessed to assimilate one carbon compound (e.g., methanol) into acetyl-CoA, leading to success in TAL production from methanol in *P. pastoris* for the first time.

## 4. Discussion

In this study, we reported the synthesis of TAL in *P. pastoris*, particularly when methanol was used as the sole carbon and energy source for the first time. By increasing the copy number of *Gh2PS* from two to six (strains PpTAL3~PpTAL8), TAL titer was gradually increased, indicating that the two-step decarboxylation reaction of one molecule acetyl-CoA and two molecules malonyl-CoA catalyzed by *Gh2PS* was rate-limiting for TAL synthesis. Afterwards, we further increased TAL production by introducing the PK pathway together with the heterologous xylose utilization pathway or the endogenous methanol utilization pathway. Our results demonstrated the synergistic effect between the PK pathway for acetyl-CoA generation and xylose or methanol assimilation for precursor supply. Our engineering strategy could be employed for the production of other acetyl-CoA derived compounds in *P. pastoris*.

Although we focused on the PK pathway as an auxiliary route for the synthesis of cytoplasmic acetyl-CoA, several other alternative acetyl-CoA producing pathways and shuttle mechanisms can be introduced and engineered in our future studies to further improve TAL production, such as pyruvate-formate lyase (PFL), acetylating acetaldehyde dehydrogenase (A-ALD), cytosolic pyruvate dehydrogenase (PDH_cyto_), pyruvate oxidase (PO)/phosphotransacetylase (PTA), acetate kinase (ACK)/phosphotransacetylase (PTA)/acetyl-CoA synthase (ACS*_SE_*^L641P^), carnitine shuttle (Cat), and citrate-oxaloacetate shuttle (Cit/ACL) [19,37]. Moreover, the recently reported Synthetic Acetyl-CoA (SACA) pathway provides a promising approach for the development of one-carbon biochemicals by a three-step enzymatic reaction from formaldehyde [38]. Two molecules of formaldehyde were first condensed into one molecule of glycolaldehyde catalyzed by glycolaldehyde synthase (GALS). Then, in the presence of inorganic phosphate, glycolaldehyde is converted into acetyl-phosphate by acetyl-phosphate synthase (ACPS). Finally, acetyl-phosphate is catalyzed by phosphotransacetylase (PTA) to synthesize acetyl-CoA. The thermodynamic and chemical driving forces of the pathway are favorable, with a low total Gibbs energy change (ΔrG’^m^) and a high maximum driving force (MDF), when compared with other known artificial one-carbon consuming pathways, making the SACA pathway theoretically feasible in vivo. The SACA pathway possesses several advantages over natural acetyl-CoA biosynthetic pathways, such as a high chemical driving force, carbon conservation, ATP independence, and the ability to operate under both aerobic and anaerobic conditions [38]. The methylotrophic yeast strain *P. pastoris*, which has a natural XuMP pathway located in the peroxisomes and endogenous highly expressed alcohol oxidase promoter *pAOX1*, is a preferred chassis for the assimilation of one-carbon compounds. The combination of the SACA pathway with the methanol utilization process provides a platform for producing acetyl-CoA from one-carbon resources, leading to the bulk production of industrial biotechnology products and the solution for the supply of biosynthetic raw materials. 

On the contrary, the yield of TAL from methanol (Figure 5) was much lower than that from glucose and xylose (Figure 2, Figure 4C and Appendix A, and Appendix A), probably due to the poor growth with methanol as the sole carbon and energy source (Figure 5). Therefore, we should focus more on metabolic engineering strategies to improve methanol tolerance (e.g., adaptive laboratory evolution) as well as methanol assimilation efficiency (genetic manipulation and protein engineering of AOX and DAS) in our future studies. Recently, adaptive laboratory evolution was employed to successfully restore cell growth in methanol with high-level production of FFA. Multi-omics analysis showed that FFA overproduction perturbed phospholipid hemostasis and the double mutations of *lpl1Δ* and *izh3Δ* played a key role in restoring phospholipid metabolism to minimize methanol toxicity [39]. The results suggested that simultaneous disruption of *lpl1* and *izh3* seems to be a promising strategy to improve methanol tolerance and methanol-based biomanufacturing efficiency in *P. pastoris*. On the other hand, considering the various advantages of the SACA pathway mentioned above, we can perform protein engineering (rational design or directed evolution) of the related enzymes GALS and ACPS to facilitate efficient synthesis of acetyl-CoA from methanol. More specifically, mutations can be introduced into the amino acid sequences of GALS and ACPS enzymes via rational protein design and/or directed evolution, thus changing the structure and activity of target proteins and ultimately improving the methanol assimilation rate.

## 5. Conclusions

Overall, this work demonstrated the potential of *P. pastoris* as a platform cell factory to synthesize acetyl-CoA derivatives from xylose and methanol for the first time. By introducing the posttranslational regulation-free *ScACC1* mutant gene and increasing the copy number of *Gh2PS*, the resultant strain PpTAL8 was able to synthesize TAL with a titer of ~2.7 g/L in YPD medium using shake flask fermentation. Meanwhile, by introducing the PK pathway and increasing the supply of Xu5P through the heterologous xylose utilization pathway or endogenous methanol utilization pathway, the production of TAL was further increased in the PpTAL11 strain, resulting in the production of 825.6 mg/L and 57.1 mg/L TAL in SCX and SCM, respectively. This is the first report on TAL biosynthesis in *P. pastoris* and its direct synthesis from methanol. Our strategy to develop an engineered *P. pastoris* strain for efficient supply of acetyl-CoA has the potential to produce a wide range of value-added compounds, such as polyketide, isoprenoid, and fatty acid derived products.

## Figures and Tables

**Figure 1 jof-09-00494-f001:**
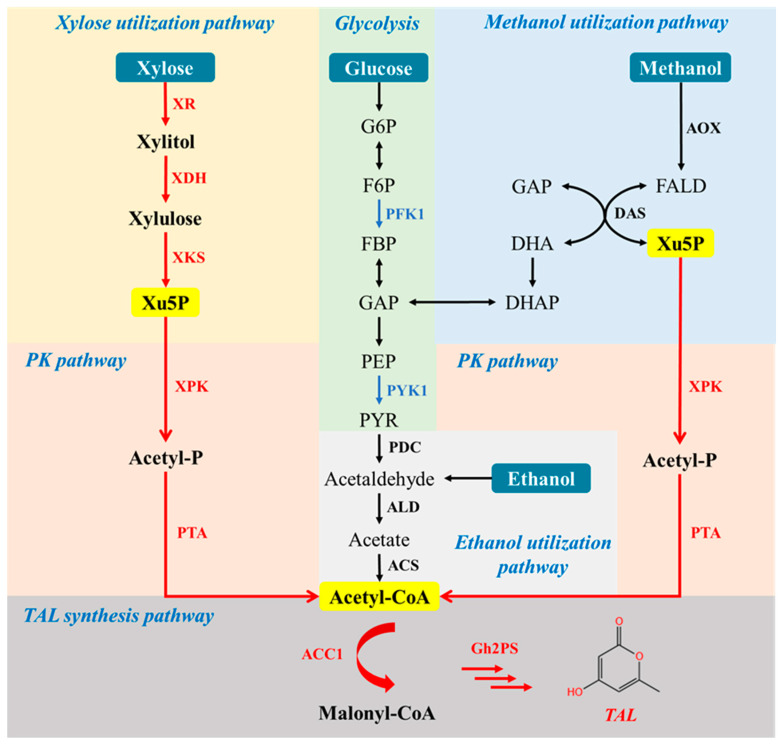
Metabolic pathway engineering for TAL synthesis in *P. pastoris***.** Heterologous genes and deleted genes are presented in red and blue, respectively. PFK1, phosphofructokinase; PYK1, pyruvate kinase; XR, xylose reductase; XDH, xylitol dehydrogenase; XKS, xylulokinase; xPK, xylulose-5-phosphate specific phosphoketolase; PTA, phosphotransacetylase; ACC1, acetyl-CoA-carboxylase; Gh2PS, 2-pyrone synthase; G6P, glucose-6-phosphate; GL6P, gluconolactone-6-phosphate; 6PG, gluconate-6-phosphate; Ru5P, ribulose-5-phosphate; Xu5P, xylulose-5-phosphate; S7P, sedoheptulose-7-phosphate; GAP, glyceraldehyde-3-phosphate; F6P, fructose-6-phosphate; E4P, erythrose-4-phosphate; FBP, D-fructose-1,6-diphosphate; PEP, phosphoenolpyruvate; PYR, pyruvate; DHA, dihydroxyacetone; DHAP, dihydroxyacetone phosphate; FALD, formaldehyde; Acetyl-P, acetyl-phosphate; TAL, triacetic acid lactone.

**Figure 2 jof-09-00494-f002:**
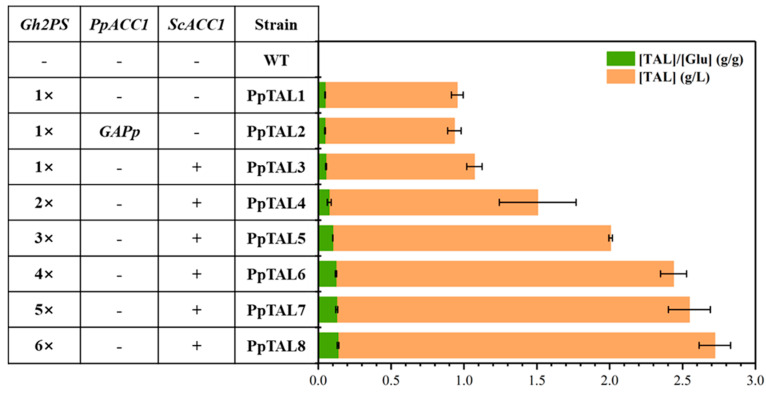
Improving TAL production in *P. pastoris* through metabolic engineering. TAL titer was improved by introducing the posttranslational regulation-free acetyl-CoA carboxylase mutant encoding gene from *S. cerevisiae* (*ScACC1**) and increasing the copy number of *Gh2PS*. All strains were fermented in YPD medium. The data represent three biological replicates, and the error bars represent standard deviations.

**Figure 3 jof-09-00494-f003:**
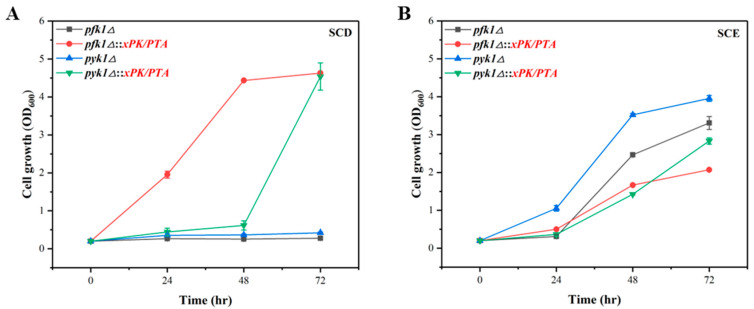
Verification of PK pathway in P. pastoris via growth complementation. Cell growth curves of pfk1Δ and pyk1Δ strains as well as the PK pathway integrated strains pfk1Δ::xPK/PTA and pyk1Δ::xPK/PTA were measured in SCD medium (**A**) and SCE medium (**B**). The data represent three biological replicates, and the error bars represent standard deviations.

**Figure 4 jof-09-00494-f004:**
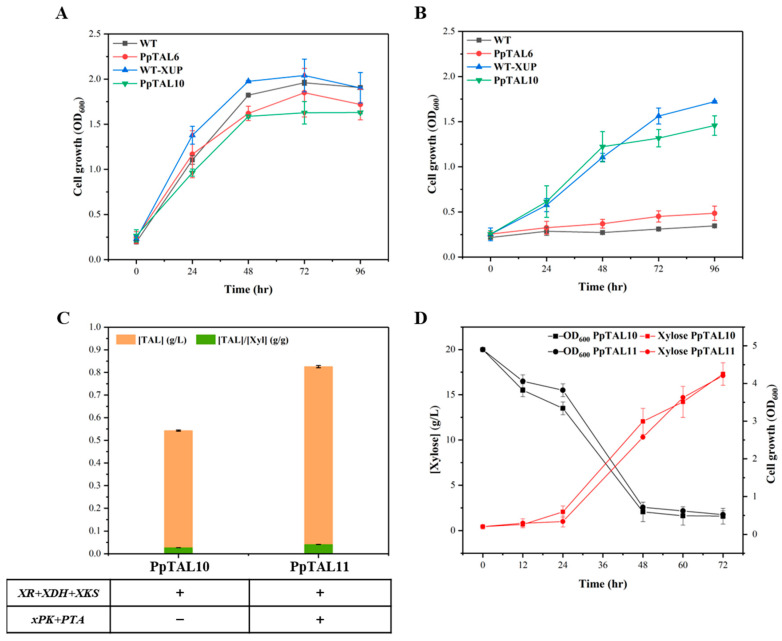
TAL production from xylose in *P. pastoris*. Functional verification of the xylose utilization pathway was performed by measuring the cell growth of WT, PpTAL6, and the corresponding PK pathway integrated strains (WT−XUP and PpTAL10) in SCX medium (**A**) and SCD medium (**B**). (**C**) Comparison of TAL titer in the xylose utilization pathway integrated strains with (PpTAL11) or without (PpTAL10) the PK pathway in SCX fermentation medium. (**D**) The xylose consumption rate and cell growth curves of PpTAL10 and PpTAL11 in SCX medium. The data represent three biological replicates, and the error bars represent standard deviations.

**Figure 5 jof-09-00494-f005:**
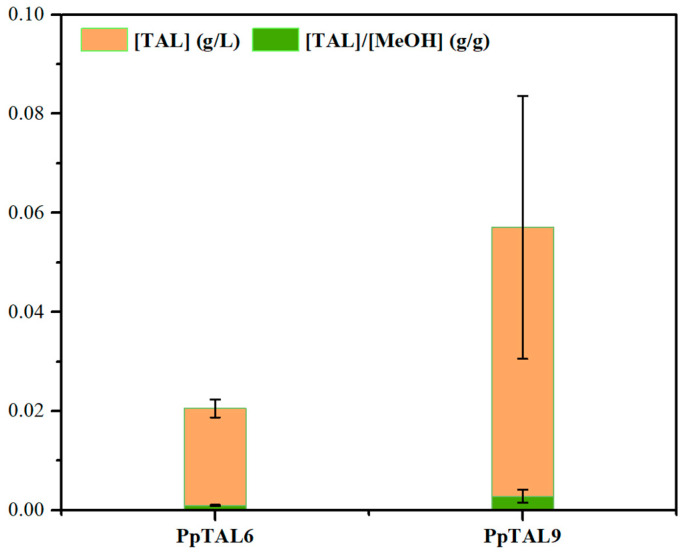
TAL production from methanol in *P. pastoris*. Comparison of TAL titer of PpTAL6 and PpTAL9 in SCM medium. MeOH, methanol. The data represent three biological replicates, and the error bars represent standard deviations.

**Table 1 jof-09-00494-t001:** Strains used in this study.

Strain	Genotype	Characteristic	Source
GS115-*Cas9*	*P. pastoris* GS115-*his4*::*Cas9*	Parent strain	[9]
PpTAL1	GS115-*Cas9* Int11::*pTEF1-Gh2PS-t0547*	One copy of *Gh2PS*	This study
PpTAL2	PpTAL1 *pACC1*::*pGAP*	*ACC1* promoter replaced with *pGAP* based on PpTAL1	This study
PpTAL3	PpTAL1 Int39::*pTEF1-ScACC1*-tAOX1*	1 copy of *Gh2PS* and 1 copy of *ScACC1**	This study
PpTAL4	PpTAL3 Int32::*pTEF1-Gh2PS-t0547*	2 copies of *Gh2PS* and 1 copy of *ScACC1**	This study
PpTAL5	PpTAL4 Int33::*pTEF1-Gh2PS-t0547*	3 copies of *Gh2PS* and 1 copy of *ScACC1**	This study
PpTAL6	PpTAL5 Int34::*pTEF1-Gh2PS-t0547*	4 copies of *Gh2PS* and 1 copy of *ScACC1**	This study
PpTAL7	PpTAL6 Int1::*pTEF1-Gh2PS-t0547*	5 copies of *Gh2PS* and 1 copy of *ScACC1**	This study
PpTAL8	PpTAL7 Int20::*pTEF1-Gh2PS-t0547*	6 copies of *Gh2PS* and 1 copy of *ScACC1**	This study
PpTAL9	PpTAL6 Int35::*pGAP-xPK-tAOX1* Int59::*pTEF1-PTA-t0547*	4 copies of *Gh2PS*, 1 copy of *ScACC1**, and 1 copy of *xPK* and *PTA*	This study
PpTAL10	PpTAL6 Int56::*pGAP-XR-tAOX1-pTEF1-XDH-t0547* Int1::*pTEF1-XKS-tAOX1*	4 copies of *Gh2PS*, 1 copy of *ScACC1**, and 1 copy of *XR*, *XDH*, and *XKS*	This study
PpTAL11	PpTAL10 Int35::*pGAP-xPK-tAOX1* Int59::*pTEF1-PTA-t0547*	4 copies of *Gh2PS*, 1 copy of *ScACC1**, 1 copy of *XR*, *XDH*, and *XKS*, and 1 copy of *xPK* and *PTA*	This study
*pfk1* *△*	GS115-*Cas9 pfk1△*	GS115-*Cas9* strain with *pfk1* deletion	This study
*pyk1* *△*	GS115-*Cas9 pyk1△*	GS115-*Cas9* strain with *pyk1* deletion	This study
*pfk1△*::*xPK/PTA*	GS115-*Cas9 pfk1△*::*xPK/PTA*	GS115-*Cas9*strain with *pfk1* deletion and 1 copy of *xPK* and *PTA*	This study
*pyk1△*::*xPK/PTA*	GS115-*Cas9 pyk1△*::*xPK/PTA*	GS115-*Cas9* strain with *pyk1* deletion and 1 copy of *xPK* and *PTA*	This study
WT-XUP	GS115-*Cas9* Int56::*pGAP-XR-tAOX1-pTEF1-XDH-t0547* Int1::*pTEF1-XKS-tAOX1*	GS115-*Cas9* strain with 1 copy of *XR*, *XDH*, and *XKS*	This study

*: ACC1 mutant (Ser659A and Ser1157A) from *S. cerevisiae* to minimize SNF1-mediated protein degradation.

## Data Availability

All relevant data are provided in the manuscript and Appendix A.

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
