# Peer review of "Metabolic Engineering of Pichia pastoris for the Production of Triacetic Acid Lactone"

_jof, 2023, doi:10.3390/jof9040494_

Round 1
Reviewer 1 Report
The manuscript by Feng and co-authors presents the production of Triacetic Acid Lactone from different carbon sources by engineered Pichia pastoris strains.
General comments:
a) A scheme representing the genomic recombination strategy would be ideal. Provide more details about the recombination cassette, i.e. homology arms, gene of interest (copy number, promoter, terminator), and selection gene, etc. This scheme is necessary for the deletion and insertion strategies.
b) The used carbon sources are interesting candidates from a biorefinery perspective, but more information is needed in the introduction and discussion section. The obtained yields (g/g) can be used to compare the strains and to estimate the production costs in an upscaled bioprocess.
c) It is not clear from your discussion which combination of carbon source and engineered strain led to the higher yields (g/g) and volumetric productivity (g/L.h). Please include this in the text (+graph or table). Calculate volumetric productivity values, if possible.
Specific comments:
L 29-31. The obtained yield (grams of TAL per gram of xylose) should be included in this portion of the abstract.
L 72-74. From the same quantity of carbon source? It would be more adequate to compare the yields (g/g), since different quantities of carbon source, or even different carbon sources, could've been used.
L 109 and Figure 1. It is not clear why the genes in blue were deleted, please elaborate. I suggest to add a cross to the deleted reactions in the pathway.
L 135-136. Please add the strain source and a reference.
L 154-162. Were all these genes from different species also subject to codon-optimization? Please clarify.
L 168. Please provide a recombination cassette scheme, and a recombination strategy scheme. It is not clear how the genomic insertion was performed and selected.
L 176. The reader would benefit from a list of characteristics of each strain. For example, strain harboring xylose reductase encoding sequence from Sc. spitipis, and so on.
L 217-219. It is not clear in which genomic loci the genes were introduced.
L 240-243. The strategy to introduce more than one copy is not clear. Please elaborate on the methodology section.
L 248-249. What is the carbon source/culture media used in this experiment? Please add all the necessary information for full understanding to each figure caption.
L 248 - Figure 2. Add calculate and add the obtained yields from the supplied carbon source. If using a complex media, provide an estimated value.
L 266-271. Were the encoding sequences codon optimized for the host strain?
L 268-271. It would be important to determine whether the genes were in fact expressed in the engineered strain. If possible, I suggest to run mRNA analysis.
L 280-282. Please add the knock-out strategy details to the methodology section.
L 282. SCD medium is not listed nor described in the methodology section. Please provide complete information in the figure caption.
L 285-288. Why did the pfk1Δ strains were able to grow on SCE medium?
L 307-310. It is not clear where the genes were introduced. Please add details.
L 310. Please clarify why it is industrially relevant to use a synthetic medium for TAL production. Is it cheaper?
L 312. SCD = SCX in your study? Please clarify.
L 333. Again, it is not immediately clear the genetic difference of GTAL10 and GTAL11. A brief explanation would help.
L 354-357. Please add the yield (g/g) information and compare to the maximum theoretical yield value from methanol.
L 386-394. Would the introduction of the proposed metabolic steps affect NAD/NADH or NADP/NADPH balance?
L 415-420. Can you estimate the production costs of TAL produced from methanol and compare with other carbon sources? Is the production from methanol energetically or economically interesting?
L 424-426. The use of methanol resistant yeast is not an option? Please clarify.
L 439-442. Engineer proteins related to methanol assimilation? Not clear, please clarify.
Reviewer 2 Report
The presented article “Metabolic Engineering of Pichia pastoris for the Production of Triacetic Acid Lactone”is devoted to the creation of a yeast strain - producer of triacetic acid lactone (TAL), which can be widely used in biotechnology. To obtain the strain of interest, the authors used not only classical approaches and methods of genetic engineering, but also metabolic engineering and CRISPR/Cas9 technology.
Using xylose metabolism genes from different organisms, the authors succeeded in obtaining a yeast strain with a heterologous xylose utilization pathway, increasing the levels of xylose-5-phosphate and acetyl-CoA in the cells of the producer strain, and demonstrating the possibility of TAL synthesis from methanol and xylose.
The results obtained will undoubtedly be of interest and useful to researchers working on the creation of yeast strains producing target compounds.
One question:
1. How stable are strain GTAL8 with six copy number of Gh2PS? Have you observed the loss of some copies due to recombination?

Round 2
Reviewer 1 Report
Manuscript quality improved.